# A Starch- and Sucrose-Reduced Diet Has Similar Efficiency as Low FODMAP in IBS—A Randomized Non-Inferiority Study

**DOI:** 10.3390/nu16173039

**Published:** 2024-09-09

**Authors:** Bodil Roth, Mohamed Nseir, Håkan Jeppsson, Mauro D’Amato, Kristina Sundquist, Bodil Ohlsson

**Affiliations:** 1Department of Clinical Sciences, Lund University, 221 00 Lund, Sweden; bodil.roth@med.lu.se (B.R.); mo1138ns-s@student.lu.se (M.N.); kristina.sundquist@med.lu.se (K.S.); 2Department of Internal Medicine, Skåne University Hospital, 205 02 Malmö, Sweden; 3Department of Clinical Nutrition, Skåne University Hospital, 221 85 Lund, Sweden; hakan.jeppsson@skane.se; 4Department of Medicine and Surgery, LUM University, 70010 Casamassima, Italy; mdamato@cicbiogune.es; 5Gastrointestinal Genetics Lab, CIC bioGUNE—BRTA, 48160 Derio, Spain; 6Ikerbasque, Basque Foundation for Science, 48007 Bilbao, Spain; 7University Clinic Primary Care Skåne, 202 13 Malmö, Region Skåne, Sweden

**Keywords:** dietary intervention, irritable bowel syndrome, low FODMAP, randomized trial, SSRD

## Abstract

A diet with low content of fermentable oligo-, di-, and monosaccharides and polyols (FODMAP) is established treatment for irritable bowel syndrome (IBS), with well-documented efficiency. A starch- and sucrose-reduced diet (SSRD) has shown similar promising effects. The primary aim of this randomized, non-inferiority study was to test SSRD against low FODMAP and compare the responder rates (RR = ∆Total IBS-SSS ≥ −50) to a 4-week dietary intervention of either diet. Secondary aims were to estimate responders of ≥100 score and 50% reduction; effects on extraintestinal symptoms; saturation; sugar craving; anthropometric parameters; and blood pressure. 155 IBS patients were randomized to SSRD (n = 77) or low FODMAP (n = 78) for 4 weeks, with a follow-up 5 months later without food restrictions. The questionnaires Rome IV, IBS-severity scoring system (IBS-SSS), and visual analog scale for IBS (VAS-IBS) were completed at baseline and after 2 and 4 weeks and 6 months. Weight, height, waist circumference, and blood pressures were measured. Comparisons were made within the groups and between changes in the two groups. There were no differences between groups at baseline. The responder rate of SSRD was non-inferior compared with low FODMAPs at week 2 (79.2% vs. 73.1%; *p* = 0.661;95% confidence interval (CI) = −20–7.2) and week 4 (79.2% vs. 78.2%; *p* = 1.000;95%CI = −14–12). Responder rate was still high when defined stricter. All gastrointestinal and extraintestinal symptoms were equally improved (*p* < 0.001 in most variables). SSRD rendered greater reductions in weight (*p* = 0.006), body mass index (BMI) (*p* = 0.005), and sugar craving (*p* = 0.05), whereas waist circumference and blood pressure were equally decreased. Weight and BMI were regained at follow-up. In the SSRD group, responders at 6 months still had lowered weight (−0.7 (−2.5–0.1) vs. 0.2 (−0.7–2.2) kg; *p* = 0.005) and BMI (−0.25 (−0.85–0.03) vs. 0.07 (−0.35–0.77) kg/m^2^; *p* = 0.009) compared with baseline in contrast to non-responders. Those who had tested both diets preferred SSRD (*p* = 0.032). In conclusion, a 4-week SSRD intervention was non-inferior to low FODMAP regarding responder rates of gastrointestinal IBS symptoms. Furthermore, strong reductions of extraintestinal symptoms were found in both groups, whereas reductions in weight, BMI, and sugar craving were most pronounced following SSRD.

## 1. Introduction

Irritable bowel syndrome (IBS) is the most common disorder of gut-brain interaction (DGBI) with a global prevalence of 4% according to the Rome IV criteria [1,2]. Functional gastrointestinal (GI) symptoms are observed in 40% of the general population [2]. Most patients experience aggravated GI symptoms after food intake [3]. The established treatment of IBS is dietary modification with the National Institute for Health and Care Excellence (NICE) guidelines and low content of fermentable oligo-, di-, and monosaccharides and polyols (FODMAP) [4,5]. Low FODMAP focus on reduction of lactose, fructose more than glucose, fructans, galacto-oligosaccharides, and polyols [6]. Fermentable carbohydrates lead to increased small intestinal water content and colonic gas production, rendering abdominal distention and altered bowel habits [7]. Since IBS is associated with visceral hypersensitivity [8], abdominal distention in these patients may lead to symptoms such as bloating and abdominal pain [7]. However, some patients complain about the complicated low FODMAP advice, and the challenge to personalize the diet [9]. Reduction of several food items may lead to malnutrition in the long-term [10], although studies have shown good nutritional supply by low FODMAP [11]. Furthermore, 25–50% of IBS patients still experience symptoms on a low FODMAP diet [5]. Thus, there is a need for other treatment options as well in IBS.

Congenital sucrase-isomaltase deficiency (CSID) is considered a rare disease [12]. During the last years, increased prevalence of rare variants of sucrase-isomaltase (*SI*) genes in IBS patients has been found and raised the hypothesis that a subgroup of IBS may represent enzymatic SI deficiency [12,13,14]. When ingested sugars are not hydrolyzed and absorbed but accumulated in the bowel, the osmotic load is increased leading to retention and secretion of water and electrolytes into the lumen. In colon, the undigested sugars lead to fermentation. Thus, symptoms like flatulence, bloating, abdominal pain, and altered bowel habits are evoked [15]. The treatment of SI deficiency is reduced intake of starch and sucrose [16]. Supplementation of sacrosidase may be considered, but starch reduction is still necessary [12,17]. The high consumption of sugar and processed food in the Western world may enhance the risk to develop symptoms by genetic variants of enzyme deficiency [13,14,18,19].

Due to the recent findings [13,14], the efficacy of a starch- and sucrose-reduced diet (SSRD) has been examined in IBS, with marked improvement of symptoms [20,21]. SSRD focus on reduction of sucrose, starch, and added sugar [22]. Although some overlaps between the diets, the largest differences between low FODMAP and SSRD is that SSRD allows intake of fructan, fructose, lactose, and sweeteners, whereas low FODMAP is less restricted regarding sucrose and starch [5,23]. The primary aim of the present non-inferiority study was to test SSRD against the established low FODMAP and compare the responder rates (RR = ∆Total IBS-SSS ≥−50) to a 4-week dietary intervention of either diet. Secondary aims were to estimate the responder rates after 6 months and with score reductions of ≥100 or ≥50% in IBS-SSS from baseline; effects on extraintestinal symptoms; saturation; sugar craving; anthropometric parameters; and blood pressure.

## 2. Materials and Methods

### 2.1. Study Design

An open randomized, non-inferior trial, with two parallel groups, was conducted at the Department of Internal Medicine, Skåne University Hospital. Malmö, Sweden, between March 2022–February 2024 [24]. After a 10-day run-in period (baseline), a thorough physical examination was performed including auscultation of heart and lungs. A dietary intervention of either SSRD or low FODMAP was given for 4 weeks. When completing the intervention, participants received information about the diet not randomized to, which they were free to test, and were followed up 5 months later without any mandatory dietary restrictions during this time. However, participants in the low FODMAP group had to reintroduce FODMAP-containing food again, one by one, according to clinical routines [4,5]. The study questionnaire, food diary, Rome IV [25], irritable bowel syndrome-severity scoring system (IBS-SSS) [26], and visual analog scale for irritable bowel syndrome (VAS-IBS) [27] were completed and abdominal palpation, weight, height, waist circumference, and blood pressure were measured at all three physical visits. IBS-SSS and VAS-IBS were also completed at home after 2 weeks of intervention.

### 2.2. Patients

Patients with a diagnosis of IBS according to Rome IV [1] and age 18–70 years with symptoms >175 scores on IBS-SSS [26], and without alcohol and/or drug abuses, current eating disturbances, pregnancy, presence of any organic GI disease, severe GI surgery in the past, severe organic and psychiatric diseases, severe food allergy, or on gluten-free-, vegan-, low FODMAP-, or low-carbohydrate high-fat (LCHF) diets were recruited to the study (Appendix A). Celiac disease was excluded by measuring transglutaminase antibodies at the Department of Clinical Chemistry [28].

A detailed description of the recruitment process has been published [24]. Briefly, a data search was performed from medical records in the County of Region Skane using the International Classification of Diseases (ICD) revision 10 for patients who had received any of the diagnoses K58.1 (diarrhea-predominated IBS; IBS-D), K58.2 (constipation-predominated IBS; IBS-C), K58.3 (mixed IBS; IBS-M), and K58.8 (other and unspecified IBS; IBS-U) during 2019–2022. Of these, 744 were randomly contacted by letter and phone call. Information letters with leaflets for distribution to waiting rooms were sent to 203 primary healthcare centers (PCC) in the County and several lectures were held for healthcare staff. Campaigns in social media were performed by a professional company (Trialy, Gothenburg, Sweden), to recruit persons with a known diagnosis of IBS. From 300 eligible patients, 214 patients were randomized to either SSRD or low FODMAP according to block randomization (BR), and 155 (72.4% of randomized cases) could enter the dietary intervention since many did not come to first visit or did not fulfill inclusion criteria (Appendix A) (BO). This means an inclusion rate of 42.7% in the group from social media and 6.5% in the group from medical records. Seven of the included IBS patients had total IBS-SSS just below 175 but were included due to clear diagnosis of IBS [1].

### 2.3. Dietary Advice

Verbal and written dietary advice were given at the first visit. Patients randomized to SSRD focused on starch- and sucrose reduction, and increased intake of certain fruits and vegetables, meat, fish, and dairy products. The dietary advice was modified from dietary guidelines for patients with CSID [22], and previously described in detail [23]. Briefly, all sucrose-containing foods were to be avoided. One serving per day was allowed for (1) whole-grain bread or oatmeal porridge and (2) fiber-rich alternatives of rice or pasta. For those not tolerating fibers, smaller amounts of regular processed rice and pasta was allowed. Whole grains were recommended instead of processed breakfast cereals. Pork, beef, lamb, fish, turkey, chicken, and egg could be ingested without any restrictions. Processed meat such as bacon, sausage, and pies should be avoided if containing sugar or starch. Natural dairy products, but not oat milk and soya milk, were allowed. Butter and oil intake was unrestricted, but margarine should be avoided. Salt, pepper, and fresh herbs could be used unrestrictedly. Nuts and seeds were recommended in place of sugary snacks. Increased fat and/or protein intake and prolonged chewing was encouraged, to enhance salivary amylase breakdown of starch and to delay GI transport. Patients were provided lists of suitable fruits and vegetables with less starch content (Appendix A). No re-introduction procedure was considered after SSRD treatment.

Participants randomized to the low FODMAP received information how to avoid or reduce intake of fructans (e.g., wheat, onion, garlic), galacto-oligosaccharides (e.g., pulses), lactose (e.g., milk), fructose more than glucose (e.g., honey), and polyols (e.g., apples, pears) during the 4-week intervention [6]. After the 4 weeks, they had to reintroduce FODMAP-containing food again, one by one, to finally find their personalized form of food, which is the routine procedure in the low FODMAP strategy, since the diet excludes several food items [10,29].

Both groups received recipes and menu suggestions to enable compliance to the diet. All participants had to continue with their ordinary energy intake, degree of physical activity, medications, probiotics, or supplements, without making any changes or introductions of new drugs or other diets. No advice was given regarding food intake frequency or regularity. The participants could reach the study staff by telephone or email throughout the study.

### 2.4. Questionnaires

#### 2.4.1. Study Questionnaire

All study participants completed a questionnaire regarding sociodemographic factors, lifestyle habits, pregnancies and childbirth, medical history, drug treatments, and family history. Two 100 mm visual analog scales (VAS) were used to estimate saturation and sugar craving, respectively.

Food intake was registered digitally for 3 days (Wednesday-Friday) at baseline, and at week 4 and month 6 at the platform called Riksmaten Flex 2021 of the Swedish Food Agency [30].

Screening for symptoms of eating disorders was performed by the SCOFF (Sick, Control, One, Fat, Food) screening tool, which was also completed after 4 weeks and at follow-up [31].

#### 2.4.2. Rome IV Questionnaire

Questions No 40–48 in the Swedish version of the Rome IV questionnaire was used, after having received license from The Rome Foundation, Inc. Raleigh, NC, USA [25].

#### 2.4.3. Irritable Bowel Syndrome-Severity Scoring System

Abdominal pain, abdominal distension, satisfaction with bowel habits, and the impact of bowel habits on daily life was estimated using VAS ranging from absent (0 mm) to very severe (100 mm) symptoms, and days with abdominal pain the last 10 days was reported. The maximum achievable score is 500. Scores ranging 75–174 indicate mild disease, 175–299 indicate moderate disease, and ≥300 indicate severe disease. Extraintestinal symptoms (nausea, difficulties to eat a whole meal, reflux, belching/excess wind, headache, back pain, leg pain, muscle/joint pain, urinary urgency, and fatigue) were estimated on VAS scales with maximal achievable score of 500 [26].

#### 2.4.4. Visual Analog Scale for Irritable Bowel Syndrome

The VAS-IBS covers the symptoms abdominal pain, diarrhea, constipation, bloating and flatulence, vomiting and nausea, psychological well-being, and intestinal symptoms’ influence on daily life, ranging from absent (0 mm) to very severe (100 mm) symptoms. The values are inverted from the original version [27], and reference values for healthy controls are defined [32].

### 2.5. Statistical Analyses

Power calculation was based on non-inferiority where SSRD was tested against a standard treatment (low FODMAP). Primary outcome was responder rate (RR = ∆Total IBS-SSS ≥ −50) and was assumed to be 65% in both treatment groups at week 4. A difference in responder rate as large as 20% in favor of the standard treatment, would allow SSRD to be non-inferior. Sample size based on 80% power, a one-sided confidence level of 97.5% and an expected loss of follow-up of 10% to confirm non-inferiority was calculated to be 100 patients in each group. Due to few dropouts, the study was closed after inclusion of 155 patients, after a second consultation with the statistician.

The statistical calculations were performed as intention-to-treat in IBM SPSS, version 29. Two participants had acute gastroenteritis at the end of the 4-week period, why the questionnaires reflecting GI symptoms from the 2-week follow-up was used. According to the Kolmogorov-Smirnov test, anthropometric and symptom data were not normally distributed and therefore presented as median (interquartile ranges) or number (percentages). Mann-Whitney U-test, Wilcoxon Signed Ranks, and Spearman’s correlation test was used for continuous variables and Fisher’s exact test was used for dichotomous variables, as well as Proportionality test for 95% confidence interval (CI). To adjust for multiple comparisons in the multiple comparison with baseline and correlations, crude *p*-values as well as the *p*-values adjusted for false discovery rate (FDR) set at 5% according to the Benjamin-Hochberg method were performed [33]. Normally distributed data in nutritional variables were presented as mean ± standard deviation (SD) and calculated by Independent-Sample or Paired-Samples *t*-test. One sample t-test were used for proportional differences. ANOVA with Bonferroni correction was performed to calculate response differences due to IBS subgroups. *p* ≤ 0.05 was considered statistically significant.

## 3. Results

### 3.1. Basal Characteristics

In total, 155 patients were included in the study categorized as IBS-C (n = 26, 16.8%), IBS-D (n = 44, 28.4%), IBS-M (n = 54, 34.8%), IBS-U (n = 7, 4.5%), and unspecific functional bowel disorder (FBD) (n = 24, 15.5%) with weekly abdominal pain but weak (<30%) association between the pain and bowel habits. Seventy-seven participants were randomized to SSRD of which 72 (93.5%) completed the 4-week intervention in comparison to 72 (92.3%) of the 78 participants randomized to low FODMAP (*p* = 1.00). Fifty-three in the SSRD group (68.8%) and 49 (62.8%) in the low FODMAP group completed the 6-month follow-up (*p* = 0.499). There was no difference in sex distribution, age, weight, body mass index (BMI), sociodemographic factors, or lifestyle habits between groups (Table 1).

Overweight/obesity was observed in 50.3% of the participants. The most common comorbidities were eczema (n = 19, 12.3%), allergy (n = 17, 11.0%), and reflux/hiatus hernia (n = 18, 11.6%). The most used drugs were paracetamol (n = 54, 34.8%), proton pump inhibitors (n = 48, 31.0%), allergy medicines (n = 24, 15.5%), and hormonal anticonception (n = 24, 15.5%). As many as 55 participants (35.5%) were using any nutritional supply in the form of minerals and vitamins (Appendix A). At inclusion, 19 (12.3%) had gluten-reduced diet, 63 (40.6%) lactose-free diet, and 14 (9%) were vegetarians. Altogether, 78 (50.3%) were already on any food-restriction diet. Twenty-eight (18.1%) participants had a history of any eating disorders.

### 3.2. Gastrointestinal and Extraintestinal Symptoms

The compliance to diet seemed good, with reductions of energy, carbohydrates, sucrose, and monosaccharides in both groups, whereas starch, disaccharides, added sugar, and alcohol only were decreased in SSRD group. Protein intake was increased in the SSRD group (*p* = 0.002), whereas fiber intake was reduced (*p* = 0.001), and alcohol intake was increased at follow-up (*p* < 0.001), in the low FODMAP group. The reductions were most pronounced in the SSRD group for carbohydrates, sucrose, starch, disaccharides, and added sugar (*p* < 0.001 for all), with differences also in fat intake between groups (*p* = 0.007) (Appendix A). The vast majority were responders to the diet, without any differences in the responder rate between the groups when calculated as intention-to-treat at week 2 (79.2% vs. 73.1%; *p* = 0.661; 95% CI = −20–7.2) and week 4 (79.2% vs. 78.2%; *p* = 1.000; 95% CI = −14–12). At follow-up, there was no significant difference between the responder rates in the groups when calculated by Fisher’s exact test (36.4% vs. 42.3%; *p* = 0.252), but when calculating the CI according to the non-inferior intention at 4 weeks, there was a borderline significance for SSRD at follow-up (95% CI = −9.4–21). Responder rate was still high when defined stricter (∆Total IBS-SSS ≥−100), both at week 2 (63.6% vs. 52.6%), week 4 (67.5% vs. 65.4%), and month 6 (20.8% vs. 28.2%). Also, a considerable responder rate was found when defined as a 50% decrease of total IBS-SSS (Figure 1).

Calculations per protocol showed similar results (Appendix A). There were fewer responders in those with a vegetarian diet when considering >100 points and 50% reduction of total IBS-SSS (*p* = 0.016 and *p* = 0.018, respectively). When calculating all who had any diet such as gluten-reduced, lactose-free, or vegetarians, the 50% responder rate was lower compared with those without any dietary restrictions (*p* = 0.031). Responders had slightly higher total IBS-SSS at baseline compared with non-responders both considering week 2 (316 (240–352) vs. 255 (235–340); *p* = 0.274), week 4 (316 (240–352) vs. 262 (222–356); *p* = 0.428), and month 6 (322 (258–366) vs. 282 (228–345); *p* = 0.116). Those who were responders (∆Total IBS-SSS ≥−50) at 6 months in the SSRD group, were those who still had lowered weight (−0.7 (−2.5–0.1) vs. 0.2 (−0.7–2.2) kg; *p* = 0.005) and BMI (−0.25 (−0.85–0.03) vs. 0.07 (−0.35–0.77) kg/m^2^; *p* = 0.009) compared with baseline (Figure 2A). Similar results were found of 50% decrease regarding BMI (*p* = 0.027) and weight (Figure 2B).

All specific GI symptoms, as well as total IBS-SSS, decreased already after 2 weeks. The improvement remained in all symptoms except for constipation in the SSRD group (Table 2). Adjustment for multiple comparisons by FDR did not change the results (Appendix A). The IBS-D group differed from IBS-C and IBS-M in improvements of diarrhea and from IBS-M in improvements of constipation in the SSRD group at week 4, whereas IBS-C differed from IBS-D and IBS-M in improvements of diarrhea in the low FODMAP group (Appendix A). About one-quarter of the participants were without symptoms after 4 weeks (<75 scores in total IBS-SSS), and 40–50% experienced abdominal pain only 2–3 times/monthly after 6 months, thus, not fulfilling the IBS criteria (Table 3).

The total burden av extraintestinal symptoms at baseline correlated with constipation (rs = 0.260, *p* = 0.001), bloating and flatulence (rs = 0.214, *p* = 0.008), vomiting and nausea (rs = 0.366, *p* < 0.001), and impaired psychological well-being (rs = 0.243, *p* = 0.003). All extraintestinal symptoms, except leg pain in the SSRD group, were improved after the dietary intervention, which remained throughout the study period in most variables (Table 4). Adjustment for multiple comparisons by FDR did not change the results (Appendix A).

The effect on total IBS-SSS or extraintestinal IBS-SSS was not affected of gluten-reduced, lactose-free, or vegetarian diets. Categorization into age groups 18–39 years, 40–59 years, and 60–70 years, did not have any effect on responder rates of ≥50 scores reduction after SSRD or low FODMAP at week 4 (*p* = 0.679 and *p* = 0.159, respectively). There was a strong correlation between the improvements in total IBS-SSS and total extraintestinal IBS-SSS at both week 2, week 4 and month 6 (rs = 0.390, rs = 0.409, and rs = 0.439, respectively; *p* < 0.001 for all) and psychological well-being (rs = 0.272, rs = 0.309, and rs = 0.366, respectively; *p* < 0.001 for all). Fatigue (rs = 0.363 and *p* < 0.001, rs = 0.431 and *p* < 0.001, and rs = 0.204 and *p* = 0.040, respectively) and belching/excess wind (rs = 0.239 and *p* = 0.004, rs = 0.419 and *p* < 0.001, and rs = 0.324 and *p* < 0.001, respectively) showed the strongest correlations among extraintestinal symptoms. After FDR adjustment, the significant correlations disappeared between changes of total IBS-SSS and headache at week 2, and of urinary urgency and muscle/joint pain at week 4 (Appendix A).

In the SSRD group, there was a weak correlation between decreased bloating and disaccharide intake (rs = 0.245, *p* = 0.044). In the low FODMAP group, there was a weak correlation between decreased bloating and monosaccharide (rs = 0.252, *p* = 0.040) and fiber intake (rs = 0.244, *p* = 0.046), as well as decreased total IBS-SSS and monosaccharide intake (rs = 0.258, *p* = 0.034). Sugar craving was decreased, especially following SSRD, whereas the degree of saturation was unaffected (Table 5, Appendix A).

### 3.3. Anthropometric Data

Weight and BMI were decreased in both groups (*p* < 0.001) but was most pronounced following SSRD (*p* = 0.006 and *p* = 0.005, respectively). After the intervention, weight and BMI were regained, but waist circumference was still reduced after 6 months (Table 5). The significant changes remained after adjustments for multiple comparisons by FDR (Appendix A). The difference in weight and BMI from baseline at month 6 correlated with the difference of total IBS-SSS (Appendix A). The systolic blood pressure was slightly decreased, whereas the diastolic blood pressure was significantly decreased, and remained decreased in the low FODMAP group (Table 5).

### 3.4. Follow-Up

Between week 4 and month 6, participants were instructed to eat whatever they preferred. At month 6, the majority (68 out of 102) had only tested the diet they were initially randomized to. Of those 34 participants who had tested both diets, 22 (64.7%) preferred SSRD and 10 (29.4%) preferred low FODMAP (*p* = 0.032), whereas 2 participants (5.9%) experienced the diets equal. The reason for preferring either diet was: easier to follow (n = 16, 47.0%), had better effect (n = 9, 26.5%), easier to follow and better effect (n = 7, 20.6%), or other (n = 2, 5.9%).

In the SSRD group (n = 53), 28 (52.8%) had continued with a modified SSRD (see below), 16 (30.2%) had returned to their ordinary diet, and the rest ingested a variety of different diets. In the low FODMAP group (n = 49), 18 (36.7%) continued with a personalized low FODMAP, 23 (46.9%) had returned to their ordinary food habits, and the other ingested different diets. A greater portion of participants in the SSRD group planned to continue with a modified SSRD (n = 28) instead of returning to normal diet (n = 11) (*p* = 0.005). In the low FODMAP group, the distribution between a continued personalized low FODMAP (n = 19) or normal diet (n = 16) was equal (*p* = 0.619).

As a free comment, several participants mentioned how they had modified SSRD to avoidance of sucrose and restricted intake of bread, potatoes, rice, and pasta, which rendered sustained effect of the diet also at follow-up. Thus, these modifications were the easiest to adhere to, and the listed fruits, berries, and vegetables were of less importance in the long-term coherence.

### 3.5. Safety Outcomes

At baseline, high blood pressure was found in some patients, who were referred to the PCC. No adverse side effect was observed during the intervention, more than that one in the low FODMAP group had worse symptoms (Appendix A).

## 4. Discussion

The responder rate of the 4-week SSRD intervention regarding GI symptoms was not inferior to the responder rate of low FODMAP. Age did not affect responder rate, but those without any prior dietary restrictions had higher responder rate, as well as those in the SSRD group who still had lowered weight at month 6 compared with baseline. All specific GI symptoms, except constipation, were decreased throughout the study. A quarter of the participants had no symptoms after the 4-week intervention, and at follow-up 5 months later, 40–50% did not fulfill the IBS criteria but were classified as unspecific FBD or healthy. In addition, all extraintestinal symptoms, except leg pain in the SSRD group, were improved after the intervention, which remained throughout the study period in most variables. The improvements in total IBS-SSS correlated strongly with improvements in total extraintestinal IBS-SSS and psychological well-being. Fatigue and belching/excess wind showed the strongest correlations with improvement of total IBS-SSS among the extraintestinal symptoms. Sugar craving was reduced, especially after SSRD, whereas saturation was unaffected in both groups. Weight and BMI were reduced after 4 weeks, most pronounced after SSRD, but were regained at follow-up. Waist circumference was still reduced at follow-up, as was also the reduced diastolic blood pressure in the low FODMAP group. Of those who had tested both diets, most preferred SSRD since it was easier to adhere to and was experienced as more efficient.

The most pronounced dietary effect on IBS has been found in patients with higher GI symptoms [34], in line with the present findings. Extraintestinal symptoms, which correlated with specific GI symptoms, are common in IBS and may be as bothersome as the GI symptoms for the patient [35]. Therefore, it is important to measure improvements in both GI and extraintestinal symptoms, to ensure an improved quality of life.

The objective of the present study was to compare the effect on symptoms of a starch- and sucrose reduction and a low FODMAP. As found in the diary books, although low FODMAP led to reduced intake of carbohydrates and sucrose, and both groups had the same extent of monosaccharide reduction, SSRD led to markedly more reductions of carbohydrates, sucrose, starch, disaccharides, and added sugar than low FODMAP. The greatest differences between the diets are gluten (due to fructan reduction)- and lactose restrictions in low FODMAP but not in SSRD. However, there are many overlaps between current diets. There are also overlaps regarding mechanisms behind the symptoms, with attempts to reduce accumulation of carbohydrates in the bowel leading to increased amounts of gas and luminal water evoking flatulence and bloating, pain, and altered bowel habits [7,15]. Maybe it is overload of carbohydrates *per se* that is most important, since a low-carbohydrate diet with higher amounts of protein and fat was equally efficient as low FODMAP [36]. The current SSRD diet does not reduce the carbohydrate intake so dramatically as the former described low-carbohydrate diet [36]. Thereby, the fat intake was not increased. Garlic, leek, and beans should be avoided in both diets. Also, low FODMAP renders lower starch and sucrose intake, since a lot of cakes and candies contain gluten and/or lactose, leading to avoidance of similar food items as SSRD [23,29]. Furthermore, sucrose reduction leads to fructose reduction. The concept of avoiding sucrose and starch is easier to adhere to, in comparison to fermentable carbohydrates, which was reflected by the preference of SSRD both in the present and the first SSRD intervention [37]. At follow-up, those who still adhered to SSRD experienced a markedly improvement of GI symptoms, and they had found out that avoidance of sucrose and reduction of bread, potatoes, rice, and pasta was enough to remain asymptomatic. Thus, the restrictions of other fruits, berries, and vegetables should be used by caution, to avoid unnecessary reduction of important micronutrients. The reduced weight and BMI in responders at 6 months in the SSRD group suggest better compliance in responders than non-responders.

Our interventional study supports previous findings in a large population-based cohort (366,432 individuals) of the importance of human carbohydrate-active enzymes (hCAZyme) genotype in relation to IBS risk [38]. *SI* genes codes for the brush-border disaccharidase sucrase-isomaltase that break down sucrose and starch whereas *AMY1B* and *AMY2A* code for amylase that break down starch into smaller sugars and disaccharides. Hypomorphic variants of *SI*, *AMY1B*, and *AMY2A* genes have been found to be associated with increased risk of IBS [38]. Hence, reduced enzymatic activity may lead to excess carbohydrates in the bowel, leading to IBS symptoms through bacterial fermentation and osmosis [15]. In alignment, decreased intake of disaccharides in the SSRD group correlated with less bloating. Future studies are warranted to identify the individual saccharides most likely to exert therapeutic effects, eventually in relation to their breakdown capacity in individual patients (i.e., battery of hCAZymes and relative efficiency of these). The marked improvement of symptoms after SSRD cannot be explained solely due to genetic variants. Other mechanisms such as microbiota modification, endocrine effects, and fructose intolerance may be involved, although more studies are needed to confirm the importance of these components [39,40].

Several reports have shown that the symptoms in IBS patients are not explained by food allergy or food intolerances [41]. Instead, poor dietary habits with food intake that trigger GI symptoms as well as leads to micronutrient deficiency are common in IBS [37,42]. Therefore, overloading of the physiological mechanisms when ingesting Western processed food with additives of sugar and starch may be one cause of symptoms in a subset of IBS patients. Sugar has not only an adverse effect on GI symptoms but is also associated with overweight and obesity [43], leading to several diseases including metabolic diseases, pain hypersensitivity, and malignancies [44,45]. A systematic review has shown obesity to be a risk factor for IBS [46], and IBS was associated with the metabolic syndrome in a large population-based study [47]. This underlines the importance of reduced weight and blood pressure in IBS, and not only improvement of symptoms. Total sugar and fructose intake are associated with all-cause mortality and cardiovascular disease mortality [48]. Thus, decrease of carbohydrate intake, especially sucrose, is of benefit for the whole society [23]. The high prevalence of previous eating disorders in IBS is in alignment with other studies [49].

Due to the great challenge to recruit participants [24], the study was completed before 200 participants were included, which may have affected the statistical calculations. However, the study cohort is still big in comparison to several others [50,51]. The main inclusion of participants from social media led to highly dedicated participants, with higher motivation than observed in the prior study when recruitment was performed from medical records [20]. The high responder rate is in line with 83% responders by Mediterranean diet [51], and 50–80% by low FODMAP [5], with at least 50% of individuals´ experiencing symptom relief in the long term [50].

One of the strengths of the present study was the follow-up for another 5 months. The results suggest that IBS patient can be offered different diets depending on their preferences. The study has several limitations, one being that the power calculation was performed for the 4-week intervention. Thus, the high rate of dropouts at month 6, with most dropouts in the low FODMAP group, led to that an obvious non-inferiority for SSRD could not be shown in the long-term.

## 5. Conclusions

In conclusion, a 4-week SSRD intervention was non-inferior to low FODMAP regarding responder rates of gastrointestinal IBS symptoms. Furthermore, strong reductions of extraintestinal symptoms were found in both groups, whereas reductions in weight, BMI, and sugar craving were most pronounced following SSRD. Those who had tested both diets preferred SSRD, since it was easier to adhere to.

## Figures and Tables

**Figure 1 nutrients-16-03039-f001:**
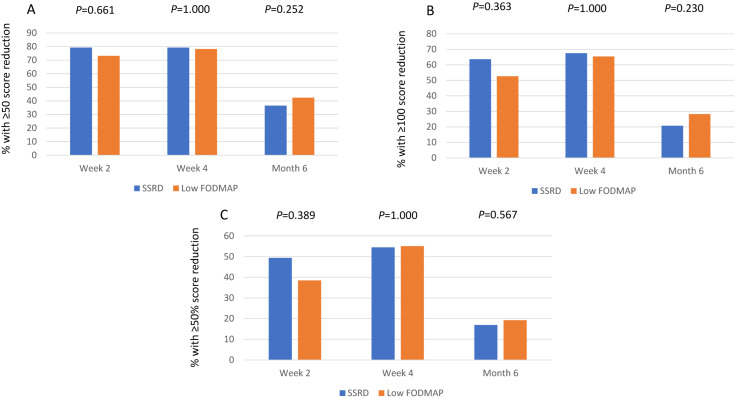
Responder rates given in percentage and calculated as intention-to-treat with (**A**) ≥50 decrease of total IBS-SSS score, (**B**) ≥100 decrease of total IBS-SSS score, and (**C**) ≥50% decrease of total IBS-SSS score. SSRD = starch- and sucrose-reduced diet with 72 patients at week 2 and 4 and 53 at month 6, out of 77. Low FODMAP = low content of fermentable oligo-, di-, and monosaccharides and polyols with 72 patients at week 2 and 4 and 49 at month 6. Fisher’s exact test. *p* ≤ 0.05 was considered statistically significant.

**Figure 2 nutrients-16-03039-f002:**
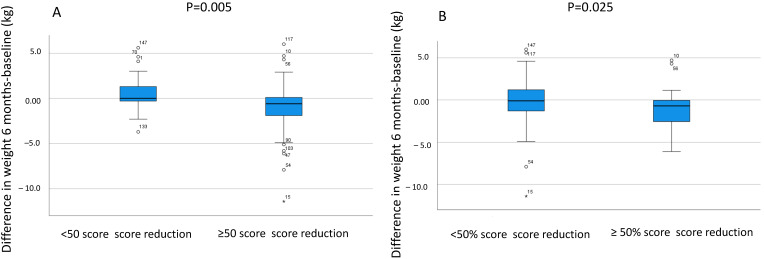
The differences in weight (kg) between 6 months and baseline in the group with starch-and sucrose-reduced diet (SSRD) divided into non-responders and responders with (**A**) ≥50 decrease of total IBS-SSS score (n = 28) and (**B**) ≥50% decrease of total IBS-SSS score (n = 13). 53 patients in the SSRD group completed the study. Median and interquartile range are given. Mann Whitney U-test. *p* ≤ 0.05 was considered statistically significant.

**Table 1 nutrients-16-03039-t001:** Basal characteristics.

Parameters	SSRD N = 77	Low FODMAP N = 78	*p*-Value
**Age (year)**	41.0 (29.5–53.0)	43.0 (33.8–56.0)	0.227
**Gender (male/female) (n,%)**	15 (19.5)/62 (80.5)	10 (12.8)/68 (87.2)	0.283
**Weight (kg)**	71.5 (63.6–82.8)	68.6 (63.0–83.4)	0.389
**BMI (kg/m^2^)**	25.1 (22.6–28.4)	24.7 (22.1–27.6)	0.479
**Disease duration (year)**	16 (7–27)	20 (10–30)	0.261
**Education (n,%)**			0.585
Primary school	5 (6.5)	2 (2.6)	
Secondary school	10 (13.0)	13 (16.7)	
Education after secondary school	20 (26.0)	17 (21.8)	
Examination at university	42 (54.5)	46 (59.0)	
**Occupation (n,%)**			0.963
Working full time	48 (62.3)	46 (59.0)	
Working 99–51%	6 (7.8)	9 (11.5)	
Working 50%	1 (1.3)	1 (1.3)	
Studying	10 (13.0)	10 (12.8)	
Sick leave	3 (3.9)	2 (2.6)	
Unemployment	3 (3.9)	2 (2.6)	
Retirement	6 (7.8)	8 (10.3)	
**Marital status (n,%)**			0.837
Living alone	16 (20.8)	14 (17.9)	
Living together	55 (71.4)	56 (71.8)	
Other	6 (7.8)	8 (10.3)	
**Smoking (n,%)**			0.086
Never	42 (54.5)	43 (55.1)	
Former	26 (33.8)	28 (35.9)	
Present un regular	2 (2.6)	6 (7.7)	
Present regular	7 (9.1)	1 (1.3)	
**Alcohol intake for 1 week (standard glass) (n,%)**			1.000
*Missing*		1	
<1	34 (44.2)	33 (42.3)	
1–4	28 (36.4)	29 (37.2)	
5–9	13 (16.9)	13 (16.7)	
10–14	1 (1.3)	2 (2.6)	
≥15	1 (1.3)	0	
**Physical activity for 1 week (n,%)**			0.475
*Missing*		1	
No time	10 (13.0)	8 (10.4)	
<30 min	11 (14.3)	14 (18.2)	
30–60 min	16 (20.8)	14 (18.2	
60–90 min	8 (10.4)	16 (20.8)	
90–120 min	8 (10.4)	8 (10.4)	
>120 min	24 (31.2)	17 (22.1)	

SSRD = starch-and sucrose-reduced diet. Low FODMAP = low content of fermentable oligo-, di-, and monosaccharides and polyols. Physical activity means activity that leads to short of breath. Values are given as number and percentages or median and interquartile values. Fisher’s exact test and Mann-Whitney U test. *p* ≤ 0.05 was considered statistically significant.

**Table 2 nutrients-16-03039-t002:** Gastrointestinal symptoms before, during and after the dietary intervention.

	SSRD N = 77			Low FODMAP N = 78			*p*-Value *
*VAS-IBS (mm)*	Value	*p*-Value	Difference	Value	*p*-Value	Difference	
**Abdominal pain** 5 (1–13)							
Baseline	47 (28–64)	-	-	50 (32–65)	-	-	0.368
2 weeks	17 (8–30)	<0.001	−27 (−47–(−9))	22 (13–40)	<0.001	−19 (−36–(−3))	0.252
4 weeks	16 (0–31)	<0.001	−24 (−44–(−9))	13 (0–27)	<0.001	−30 (−53–(−8))	0.425
6 months	32 (19–63)	0.003	−6 (−28–3)	30 (16–54)	<0.001	−16 (−38–4)	0.270
**Diarrhea** 3 (0–10)							
Baseline	53 (19–73)	-	-	37 (4–74)	-	-	0.245
2 weeks	15 (4–48)	<0.001	−14 (−51–0)	10 (0–37)	<0.001	−14 (−38–0)	0.793
4 weeks	17 (3–39)	<0.001	−8 (−48–2)	8 (0–24)	<0.001	−16 (−53–0)	0.633
6 months	31 (8–68)	<0.001	−12 (−39–0)	11 (3–44)	0.008	−8 (−30–7)	0.457
**Constipation** 6 (2–16)							
Baseline	53 (6–72)	-	-	54 (10–76)	-	-	0.439
2 weeks	16 (2–50)	<0.001	−12 (−36–2)	20 (0–68)	0.02	−6 (−28–4)	0.510
4 weeks	16 (2–43)	<0.001	−8 (−46–0)	21 (0–55)	<0.001	−13 (−33–0)	0.815
6 months	22 (0–61)	0.121	−3 (−24–12)	42 (2–70)	0.022	−4 (−32–3)	0.528
**Bloating and flatulence** 10 (2–23)							
Baseline	73 (58–88)	-	-	73 (54–86)	-	-	0.677
2 weeks	34 (18–53)	<0.001	−37 (−53–(−9))	23 (13–50)	<0.001	−39 (−56–(−16))	0.469
4 weeks	24 (10–54)	<0.001	−43 (−63–(−11)	19 (8–50)	<0.001	−44 (−61–(−25))	0.359
6 months	62 (30–75)	0.002	−15 (−38–14)	56 (33–70)	<0.001	−18 (−33–(−3))	0.416
**Vomiting and nausea** 2 (0–4)							
Baseline	13 (2–34)	-	-	13 (1–36)	-	-	0.957
2 weeks	6 (0–12)	<0.001	−7 (−20–0)	4 (0–12)	<0.001	−7 (−22–0)	0.773
4 weeks	3 (0–12)	<0.001	−6 (−15–0)	0 (0–11)	<0.001	−7 (−21–0)	0.743
6 months	8 (1–21)	0.002	−4 (−14–1)	5 (0–21)	0.009	−2 (−17–1)	0.892
**Intestinal symptom’s influence on daily life** 2 (0–14)							
Baseline	74 (57–84)	-	-	70 (54–84)	-	-	0.694
2 weeks	29 (15–60)	<0.001	−36 (−52–(−11))	30 (17–60)	<0.001	−28 (−50–(−15))	0.688
4 weeks	24 (12–62)	<0.001	−30 (−60–(−10))	22 (10–50)	<0.001	−33 (−53–(−18))	0.593
6 months	40 (23–76)	<0.001	−12 (−45–0)	48 (24–68)	<0.001	−26 (−43–(−2))	0.492
**Psychological well-being** 5 (2–15)							
Baseline	39 (15–65)	-	-	45 (16–59)	-	-	0.708
2 weeks	24 (11–42)	<0.001	−7 (−26–0)	27 (8–46)	0.021	−4 (−25–5)	0.339
4 weeks	20 (5–32)	<0.001	−12 (−29–(−2)	18 (2–34)	<0.001	−13 (−32–1)	0.788
6 months	22 (11–50)	0.009	−6 (−20–4)	26 (8–39)	0.043	−2 (−30–7)	0.911
** *IBS-SSS* **							
**Total IBS-SSS**							
Baseline	301 (233–348)	-	-	300 (238–360)	-	-	0.845
2 weeks	136 (87–223)	<0.001	−138 (−212–(−82))	151 (100–232)	<0.001	−110 (−188–(−68))	0.310
4 weeks	119 (66–230)	<0.001	−146 (−240–(−88))	116 (63–176)	<0.001	−153 (−231–(−90))	0.585
6 months	204 (146–234)	<0.001	−55 (−130–4)	220 (144–301)	<0.001	−93 (−181–(−20))	0.069

SSRD = starch-and sucrose-reduced diet with 72 patients at week 2 and 4 and 53 at month 6. Low FODMAP = low content of fermentable oligo-, di-, and monosaccharides and polyols with 71 patients at week 2 and 4 and 49 at month 6. Gastrointestinal symptoms estimated by irritable bowel syndrome -severity scoring system (IBS-SSS) [26] and visual analog scale for irritable bowel syndrome (VAS-IBS) [27]. Reference values for healthy within brackets [32]. Values are given as median and interquartile. Wilcoxon Signed Ranks for comparisons within the groups and Mann-Whitney U test * for comparison between baseline and the differences of the two groups. *p* ≤ 0.05 was considered statistically significant.

**Table 3 nutrients-16-03039-t003:** IBS categorization.

	SSRD	*p*-Value *	Low FODMAP	*p*-Value *	*p*-Value **
**Baseline**	N = 77		N = 78		0.078
IBS-C	14 (18.2)		12 (15.4)		
IBS-D	29 (37.7)		15 (19.2)		
IBS-M	22 (28.6)		32 (41.0)		
IBS-U	2 (2.6)		5 (6.4)		
FBD	10 (13.0)		14 (17.9)		
**Week 4**					0.621
IBS-C	8 (10.4)		10 (12.8)		
IBS-D	8 (10.4)		7 (9.0)		
IBS-M	14 (18.2)		9 (11.5)		
IBS-U	0		2 (2.6)		
FBD	23 (29.9)		25 (32.1)		
Healthy	19 (24.7)		19 (24.4)		
Missing	5 (6.5)	<0.001	6 (7.7)	<0.001	
**Month 6**					0.198
IBS-C	6 (7.8)		11 (14.1)		
IBS-D	8 (10.4)		6 (7.7)		
IBS-M	12 (15.6)		9 (11.5)		
IBS-U	1 (1.3)		3 (3.8)		
FBD	23 (29.9)		13 (16.7)		
Healthy	3 (3.9)		7 (9.0)		
Missing	24 (31.2)	<0.001	29 (37.2)	0.046	

SSRD = starch-and sucrose-reduced diet. Low FODMAP = low content of fermentable oligo-, di-, and monosaccharides and polyols. IBS-C = constipation-predominated IBS, IBS-D = diarrhea-predominated IBS, IBS-M = mixed IBS, IBS-U = unspecified IBS according to Rome IV questionnaire [25]. Healthy was defined as <75 in total IBS-SSS [26]. Unspecific functional bowel disorder (FBD) means at baseline the presence of abdominal pain weekly but with weak association with bowel habits. After 4 weeks and 6 months, FBD represent those with abdominal pain 2–3 times or less/month, but with total IBS-SSS > 75. Values are given as number and percentages. Fisher´s exact test. * = comparisons within the group between baseline and week 4 and month 6. ** = comparisons between the groups. *p* ≤ 0.05 was considered statistically significant.

**Table 4 nutrients-16-03039-t004:** Extraintestinal symptoms before, during and after the dietary intervention.

	SSRD N = 77			Low FODMAP N = 78			*p*-Value *
*Extraintestinal IBS-SSS*	Value	*p*-Value	Difference	Value	*p*-Value	Difference	
**Difficulties to eat a whole meal**							
Baseline	10 (2–26)	-	-	6 (0–22)	-	-	0.267
2 weeks	4 (0–12)	0.002	−3 (−13–3)	3 (0–13)	0.005	−2 (−11–0)	0.951
4 weeks	2 (0–13)	<0.001	−4 (−16–1)	0 (0–9)	<0.001	−3 (−12–0)	0.629
6 months	4 (0–14)	0.074	−2 (−13–4)	2 (0–18)	0.053	−1 (−10–0)	0.940
**Headache**							
Baseline	33 (10–66)	-	-	27 (9–58)	-	-	0.737
2 weeks	14 (5–36)	<0.001	−5 (−30–2)	15 (2–47)	<0.001	−6 (−22–0)	0.993
4 weeks	14 (2–32)	<0.001	−9 (−27–2)	12 (0–35)	<0.001	−9 (−31–0)	0.855
6 months	24 (10–55)	0.185	−1 (−21–8)	20 (4–50)	0.001	−4 (−21–2)	0.324
**Back pain**							
Baseline	20 (4–50)	-	-	28 (4–65)	-	-	0.395
2 weeks	6 (0–29)	<0.001	−6 (−21–0)	14 (0–39)	<0.001	−2 (−26–2)	0.409
4 weeks	6 (0–30)	<0.001	−7 (−22–0)	4 (0–35)	<0.001	−7 (−32–0)	0.998
6 months	23 (4–58)	0.157	−4 (−18–8)	24 (4–70)	0.675	0 (−14–12)	0.501
**Fatigue**							
Baseline	57 (30–81)	-	-	74 (48–84)	-	-	0.055
2 weeks	33 (16–68)	<0.001	−14 (−27–0)	47 (19–70)	<0.001	−12 (−32–0)	0.660
4 weeks	27 (9–56)	<0.001	−18 (−32–(−2))	37 (14–60)	<0.001	−19 (−38–(−3))	0.712
6 months	49 (18–68)	0.004	−7 (−20–4)	48 (19–69)	<0.001	−13 (−27–0)	0.128
**Belching/excess wind**							
Baseline	72 (48–85)	-	-	75 (52–87)	-	-	0.621
2 weeks	24 (10–66)	<0.001	−21 (−51––(−6))	37 (14–66)	<0.001	−23 (−44–(−6))	0.804
4 weeks	14 (6–40)	<0.001	−41 (−67–(−6))	21 (8–45)	<0.001	−41 (−59–(−19))	0.878
6 months	47 (20–68)	<0.001	−13 (−33–(−2))	48 (22–70)	<0.001	−15 (−37–(−2))	0.599
**Reflux**							
Baseline	20 (7–50)	-	-	20 (2–60)	-	-	0.678
2 weeks	5 (0–18)	<0.001	−12 (−27–0)	7 (0–35)	<0.001	−6 (−30–0)	0.274
4 weeks	4 (0–20)	<0.001	−12 (−28–0)	3 (0–26)	<0.001	−10 (−30–0)	0.945
6 months	11 (4–26)	0.002	−6 (−20–2)	21 (2–54)	0.013	−3 (−18–2)	0.727
**Urinary urgency**							
Baseline	14 (2–64)	-	-	22 (4–64)	-	-	0.491
2 weeks	7 (0–24)	<0.001	−7 (−24–0)	7 (0–33)	<0.001	−8 (−29–0)	0.598
4 weeks	4 (0–23)	<0.001	−6 (−34–0)	3 (0–22)	<0.001	−13 (−44–0)	0.290
6 months	18 (0–44)	0.003	−7 (−22–1)	16 (0–53)	0.003	−7 (−19–0)	0.975
**Leg pain**							
Baseline	2 (0–9)	-	-	0 (0–18)	-	-	0.995
2 weeks	1 (0–15)	0.281	0 (−4–2)	0 (0–10)	0.036	0 (−3–0)	0.776
4 weeks	0 (0–7)	0.024	0 (−4–0)	0 (0–5)	0.005	0 (−4–0)	0.564
6 months	2 (0–12)	0.774	0 (−2–2)	0 (0–14)	0.573	0 (−2–2)	0.682
**Muscle/joint pain**							
Baseline	25 (5–56)	-	-	30 (4–72)	-	-	0.506
2 weeks	11 (0–46)	0.002	−5 (−20–1)	18 (0–57)	0.014	−1 (−20–4)	0.616
4 weeks	13 (0–30)	<0.001	−10 (−27–0)	12 (0–39)	<0.001	−3 (−35–1)	0.699
6 months	23 (5–53)	0.032	−3 (−17–4)	19 (4–70)	0.084	−2 (−18–7)	0.755
**Total extraintestinal IBS-SSS**							
Baseline	160 (110–208)	-	-	172 (120–242)	-	-	0.268
2 weeks	96 (50–154)	<0.001	−60 (−89–(−20))	115 (55–156)	<0.001	−54 (−82–(−30))	0.852
4 weeks	91 (28–140)	<0.001	−72 (−112–(−41))	77 (44–136)	<0.001	−83 (−118–(−44))	0.408
6 months	127 (71–191)	<0.001	−44 (−75–2)	133 (78–214)	<0.001	−36 (−59–(−10))	0.977

SSRD = starch-and sucrose-reduced diet with 72 patients at week 2 and 4 and 53 at month 6. Low FODMAP = low content of fermentable oligo-, di-, and monosaccharides and polyols with 71 patients at week 2 and 4 and 53 at month 6. Symptoms estimated by irritable bowel syndrome -severity scoring system (IBS-SSS) [26]. Values are given as median and interquartile. Wilcoxon Signed Ranks for comparisons within the groups and Mann-Whitney U test * for comparison between baseline and the differences of the two groups. *p* ≤ 0.05 was considered statistically significant.

**Table 5 nutrients-16-03039-t005:** Anthropometric data.

	SSRD N = 77			Low FODMAP N = 78			*p*-Value *
Variables	Value	*p*-Value	Difference	Value	*p*-Value	Difference	
**Weight** (kg)							
Baseline	71.5 (63.6–82.8)	-	-	68.6 (63–83.4)	-	-	0.513
4 weeks	70 (63.2–81)	<0.001	−1.6 (−2.4–(−0.4))	67.8 (62.5–82.7)	<0.001	−0.8 (−1.6–(−0.1))	0.006
6 months	74.1 (66.6–85.7)	0.516	−0.2 (−1.4–1.2)	68.6 (62.8–80.8)	0.079	−0.3 (−1.6–0.6)	0.438
**BMI** (kg/m^2^)							
Baseline	25.14 (22.64–28.45)	-	-	24.68 (22.13–27.64)	-	-	0.538
4 weeks	24.8 (21.97–27.6)	<0.001	−0.55 (−0.86–(−0.15))	24.63 (22–27.32)	<0.001	−0.26 (−0.56–(−0.03))	0.005
6 months	25.95 (22.66–28.57)	0.504	−0.07 (−0.53–0.44)	25.08 (22.05–26.76)	0.089	−0.11 (−0.54–0.23)	0.526
**Waist circumference** (cm)							
Baseline	88 (76–97)	-	-	86 (79–94.8)	-	-	0.831
4 weeks	86 (75–94)	<0.001	−2 (−4–0)	85 (79–93)	<0.001	−2 (−3–1)	0.981
6 months	89 (77.5–97)	0.022	−1 (−4–1)	85.5 (80–93.5)	0.038	−1 (−3–1)	0.758
**Systolic Blood Pressure** (mmHg)							
Baseline	125 (114–139)			126 (116–139)			0.657
4 weeks	123 (114–135)	0.097	−2 (−10–6)	124 (113–135)	0.024	−3 (−8–3)	0.762
6 months	127 (116–138)	0.588	−1 (−7–6)	126 (117–136)	0.138	−3 (−12–8)	0.399
**Diastolic Blood Pressure** (mmHg)							
Baseline	81 (72–88)			81 (74–90)			0.403
4 weeks	78 (70–84)	0.006	−3 (−6–3)	80 (73–85)	<0.001	−4 (−8–2)	0.225
6 months	80 (76–86)	0.575	−2 (−6–5)	80 (72–86)	0.044	−1 (−10–3)	0.190
**Sugar craving** (mm)							
Baseline	66 (40–85)	-	-	60 (29–80)	-	-	0.384
4 weeks	34 (17–67)	<0.001	−15 (−41–0)	41 (22–69)	0.001	−8 (−23–5)	0.050
6 months	53 (31–72)	0.058	−7 (−23–10)	48 (28–72)	0.246	−2 (−10–8)	0.448
**Saturation** (mm)							
Baseline	74 (52–93)	-	-	77 (68–86)	-	-	0.833
4 weeks	83 (69–93)	0.107	4 (−12–24)	80 (62–90)	0.688	1 (−10–12)	0.261
6 months	77 (69–90)	0.473	0 (−12–16)	72 (62–87)	0.464	−2 (−13–16)	0.275

BMI = body mass index. SSRD = starch-and sucrose-reduced diet with 72 patients at week 4 and 53 at month 6. Low FODMAP = low content of fermentable oligo-, di-, and monosaccharides and polyols with 72 patients at week 4 (except for sugar craving and saturation with 71 patients) and 49 at month 6. Sugar craving and saturation estimated on visual analog scales of 100 mm, with higher scores meaning most craving and saturation. Values are given as median and interquartile. Wilcoxon Signed Ranks for comparisons within the groups and Mann-Whitney U test * for comparison between baseline and the differences of the two groups. *p* ≤ 0.05 was considered statistically significant.

## Data Availability

The data presented in this study are available on request from the corresponding author due to ethical reasons.

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
