# Peer review of "A Starch- and Sucrose-Reduced Diet Has Similar Efficiency as Low FODMAP in IBS—A Randomized Non-Inferiority Study"

_nutrients, 2024, doi:10.3390/nu16173039_

Round 1

Reviewer 1 Report

Comments and Suggestions for Authors

It is an interesting study in which authors assessed effect of both starch- and sucrose-reduced diet and low FODMAP in IBS by a randomized non-inferiority trial. Although there were some interesting findings, some technical issues should be addressed if a revision will be invited.

1.      Due to the nature of RCT, authors are suggested to clearly present primary, secondary and safety outcomes. The conclusion should focus on primary outcomes.

2.      Authors seemed not to present clearly about safety outcomes in this trial. Pls check it.

3.      4-week outcome could be main observed point, but authors compared the outcomes at 6-week. Authors said that “After the 4 weeks, they had to reintroduce food FODMAP-containing food again, one by one, to finally find their personalized form of food…” . how about group with both starch- and sucrose-reduced diet? It also continues to 6 week? If not, the comparison could be cautious.

4.      Authors said in line 236 that “Calculations per protocol showed similar results (Supplementary Figure 2).”, but actually I cannot find supplementary figure 2. Authors are suggested to check it carefully.

5.      A key point issue should be pointed out that authors seemed not to present clearly sample size during follow-up in all tables and figures. If possible, it is suggested to add them.

6.      Due to follow-up study, data are correlated during the follow-up. So repeated measure ANOVA or mixed model is suggested to use for analysis of outcome change. Otherwise, P values for each time point should be corrected.

7.      In abstract, authors said that “Reductions of weight, BMI, and sugar craving were most pronounced following SSRD.” It seems strange because this RCT focused on non-inferior of SSRD and low FODMAP. This result seems secondary one.

Author Response

Dear Editor-in-chief and Reviewers

Manuscript ID: nutrients-3166947
Type of manuscript: Article
Title: A starch- and sucrose-reduced diet has similar efficiency as low
FODMAP in IBS. A randomized non-inferiority study

We are most grateful to the reviewers for taking their time to read the manuscript and their very constructive comments. In respect to these comments, we have been able to improve the manuscript. We have tracked all changes in yellow. We hope that our work now will be regarded as suitable for publication in the Nutrients.

Bodil Ohlsson, Professors, Senior Consultant of Gastroenterology and Hepatology

Reviewer 1.

It is an interesting study in which authors assessed effect of both starch- and sucrose-reduced diet and low FODMAP in IBS by a randomized non-inferiority trial. Although there were some interesting findings, some technical issues should be addressed if a revision will be invited.

  1. Due to the nature of RCT, authors are suggested to clearly present primary, secondary and safety outcomes. The conclusion should focus on primary outcomes.

Response: Thank you for pointing this out. We agree that the focus should be focused on primary outcomes. We have more clearly stated primary and secondary aims in both abstract and introduction, page2, line 77 and 79. The conclusions have accordingly been more clearly separated in both abstract and in the first section of the discussion, page 14, line 392409, and in the conclusion, page 15, line 486-489.

  1. Authors seemed not to present clearly about safety outcomes in this trial. Pls check it.

Response: No side effects were found during the study. This is now added on page 13-14, line 387-390.

  1. 4-week outcome could be main observed point, but authors compared the outcomes at 6-week. Authors said that “After the 4 weeks, they had to reintroduce food FODMAP-containing food again, one by one, to finally find their personalized form of food…” . how about group with both starch- and sucrose-reduced diet? It also continues to 6 week? If not, the comparison could be cautious.

Response: The study continued for 6 months, not 6 weeks. It is the standard procedure that low FODMAP must be reintroduced with one food item by another during some weeks (ref No 10,29). This is because the diet makes several exclusions. This re-introduction is not necessary for the SSRD, since this diet does not exclude that many food items. This is now clarified on page 3, line 140 and line 146-147.

  1. Authors said in line 236 that “Calculations per protocol showed similar results (Supplementary Figure 2).”, but actually I cannot find supplementary figure 2. Authors are suggested to check it carefully.

Response: Thank you for pointing this out. Both Suppl Figure 1 and 2 is missing in the system. This is now uploaded in the same document together with supplementary tables and shown after the tables.

  1. A key point issue should be pointed out that authors seemed not to present clearly sample size during follow-up in all tables and figures. If possible, it is suggested to add them.

Response: Agree. Missing samples are now added in the legends to all tables and figures.

  1. Due to follow-up study, data are correlated during the follow-up. So repeated measure ANOVA or mixed model is suggested to use for analysis of outcome change. Otherwise, P values for each time point should be corrected.

Response: We have corrected for the multiple comparisons compared with baseline by false discovery rate (FDR) in Table 2, 4, and 5, explained in statistical analyses, page 5, line 200-203. These corrected values are given as supplementary tables S5, Table S7, and Table S9. This correction did not have any effect on the significance level, page 8, line 284-285, page 10, line 317-318, and page 13, line 360-361. When calculating FDR on correlations in Supplementary Table S8, all significant changes remained except for that the significant correlations disappeared between changes of total IBS-SSS and headache at week 2, and of urinary urgency and muscle/joint pain at week 4, page 12, line 337-339.

  1. In abstract, authors said that “Reductions of weight, BMI, and sugar craving were most pronounced following SSRD.” It seems strange because this RCT focused on non-inferiorof SSRD and low FODMAP. This result seems secondary one.

Response: Agree. These results are secondary aims, but we think they deserves to be published. In this revised version, the primary aim is clarified to concern the non-inferiority of SSRD in responder rate and secondary aims are the effects on extraintestinal symptoms, weight, BMI, saturation, and sugar craving. The outcomes of secondary aims are presented after the outcome of the primary aim in the abstract and in the conclusion of the discussion, page 15, line 486-489.

Reviewer 2 Report

Comments and Suggestions for Authors

The paper “A starch- and sucrose-reduced diet has similar efficiency as low FODMAP in IBS. A randomized non-inferiority study” aimed to test SSRD against low FODMAP and compare the responder rates to a 4-week dietary intervention of either diet. The research is interesting and meaningful, but there are some questions that need to be further improved or explained.

Comments:

Q1. The precision of certain expressions is inadequate. Is there an overlap between the concepts of (FODMAP) and (SSRD)? For instance, sucrose is classified as a disaccharide. In this scenario, it would be inappropriate to compare these two groups.

Q2. The labeled error lines are missing in Figure 1, and no results with significant differences are presented. While the line 209~220, some results of significant differences are demonstrated.

Q3. The expressions of results in this paper are inadequate, and the majority of the 8-12 pages consists solely of results without effective summaries, which requires supplementation. Although some of the results are addressed in the discussion section, it is insufficient.

Q4. The background of the article is insufficient. Is SSRD a known and recognized intervention for IBS? What direction did the previous FODMAP coverage presented in this article focus on? Assuming the research in this paper is rigorous, is this paper trying to prove that FODMAP is superior to SSRD? If the effects are similar, or there are no significant differences, how to reflect the significance of this study.

Q5. With regards to the action mechanism schematics of the two interventions, it is recommended that they be included in order to enhance the writing quality of this paper and facilitate comprehension.

Q6. As for the significance analysis of differences between groups, the authors need to confirm again. In Table 5, body weight and BMI, why are there significant differences between the two groups at 4 weeks, but no difference after 6 months?

Finally, as for the disscussion part in this paper, the paper lacks effective summaries of the existing literatures, and the conclusion fails to reflect the superiority of this paper.

Author Response

Dear Editor-in-chief and Reviewers

Manuscript ID: nutrients-3166947
Type of manuscript: Article
Title: A starch- and sucrose-reduced diet has similar efficiency as low
FODMAP in IBS. A randomized non-inferiority study

We are most grateful to the reviewers for taking their time to read the manuscript and their very constructive comments. In respect to these comments, we have been able to improve the manuscript. We have tracked all changes in yellow. We hope that our work now will be regarded as suitable for publication in the Nutrients.

Bodil Ohlsson, Professor, Senior Consultant of Gastroenterology and Hepatology

Reviewer 2.

The paper “A starch- and sucrose-reduced diet has similar efficiency as low FODMAP in IBS. A randomized non-inferiority study” aimed to test SSRD against low FODMAP and compare the responder rates to a 4-week dietary intervention of either diet. The research is interesting and meaningful, but there are some questions that need to be further improved or explained.

Comments:

Q1. The precision of certain expressions is inadequate. Is there an overlap between the concepts of (FODMAP) and (SSRD)? For instance, sucrose is classified as a disaccharide. In this scenario, it would be inappropriate to compare these two groups.

Response: This is an important point. Low FODMAP is the established treatment of IBS. However, the diet is rather complicated, with exclusion of several food items, with the risk of malnutrition in the long-term (Staudacher 2020). Further, many think it is too complicated to follow. Still, there are 25-50% who are not improved by low FODMAP (Mitchel 2019). We have in a previous trial found SSRD to be efficient in reduction of GI symptoms (Nilholms et al. 2019), and the patients reported that SSRD was easier to follow than the low FODMAP they had been recommended by their physician and tested before. There are several overlaps between the diets, as discussed in the discussion, but there are also some differences, page 2, line 73-77 and page 14, line 415-4323. For example, gluten and lactose are omitted in low FODMAP, but allowed in SSRD. SSRD more clearly stress to avoid sugar and processed food, which leads to less restriction of healthy food items, page 13, line 381-385 and page 14, line 432-439. Furthermore, the participants experienced SSRD to be easier to adhere to, page 13, line 368-371. Therefore, it is important to objectively examine whether there is any difference of the clinical effects between the two diets. If similar effects, the health care staff can recommend both diets, and the IBS patients may choose which diet they prefer.

In the text on page 6, line 237-242, the results of nutrient intake are shown to significantly differ between the two diets, and those results are shown in Supplementary Table S4.

Q2. The labeled error lines are missing in Figure 1, and no results with significant differences are presented. While the line 209~220, some results of significant differences are demonstrated.

Response: We do not think that error lines are needed in Figure 1, where we show the percentage of responders in each group by bar charts. In Figure 1, the results of Fisher´s exact test is shown. The results are also shown on page 6, line 242-248, where also the 95% CI is shown. At follow-up, there was a borderline-significance for inferiority of SSRD compared with low FODMAP (95% CI = -9.4-21), where the 95% CI was assumed to be ≤20 (RR=∆Total IBS-SSS ≥−50 at week 4), page 4, line 187-189. This is now further clarified in the text, page 6, line 245-248.

Q3. The expressions of results in this paper are inadequate, and the majority of the 8-12 pages consists solely of results without effective summaries, which requires supplementation. Although some of the results are addressed in the discussion section, it is insufficient.

Response: Thanks for pointing out this. The results are shown in figures and tables, and now better summarized in the beginning of the discussion, page 14, first section of the discussion.

Q4. The background of the article is insufficient. Is SSRD a known and recognized intervention for IBS? What direction did the previous FODMAP coverage presented in this article focus on? Assuming the research in this paper is rigorous, is this paper trying to prove that FODMAP is superior to SSRD? If the effects are similar, or there are no significant differences, how to reflect the significance of this study.

Response: The background to the study is now better explained in the introduction, page 2, line 49-71. Low FODMAP is the established dietary treatment of IBS, used world-wide. SSRD has only been examined in two previous trials, one performed in Sweden (Nilholm et al. 2019, ref No 20) and one in Spain (Gayaso et al. 2023, ref no 21). In the Swedish study, the IBS participants were randomized either to SSRD or to continue with their ordinary food habits. In the Spanish study, participants were their own controls, with comparison of symptoms before and after SSRD. Thus, no control group was used. This current study is the first study to compare the two diets. This study wants to examine if SSRD is non-inferior to the standard diet low FODMAP, page 4, line 185-189.

It is useful for the healthcare staff and for IBS patients, to know if there are any differences in the clinical effects, or whether the two diets are equally efficient. As stated in the response above to Q1, low FODMAP is experienced difficult to adhere to (Ref No 9), why more dietary alternatives are important to find. 

Q5. With regards to the action mechanism schematics of the two interventions, it is recommended that they be included in order to enhance the writing quality of this paper and facilitate comprehension.

Response: Low FODMAP diet focus on reduction of fermentable carbohydrates such as oligo-, di-, and monosaccharides and polyols. Therefore, foods containing these nutrients have to be avoided. SSRD focus on reduction of starch and sucrose, due to the recent finding of an association between genetic variants of sucrose-isomaltase (SI) genes, possibly representing enzymatic SI deficiency. This is now better explained in the introduction, page 2, line 49-71.

Also in the discussion, page 14, line 422-429, mechanisms are discussed. We have also found other carbohydrate-active enzymes (hCAZymes) genotypes that are related to IBS risk (Torices 2024 et al. ref No 38), page 15, line 441-449.

Q6. As for the significance analysis of differences between groups, the authors need to confirm again. In Table 5, body weight and BMI, why are there significant differences between the two groups at 4 weeks, but no difference after 6 months?

Response: During the first 4 weeks, all participants had to adhere to the respective diet. After that, during the last 5 months, they were allowed to eat whatever they preferred, page 2, line 89-92. Many of the participants then went back to their ordinary food habits, page 13, line 373-377. That may be the explanation to the regaining of weight.

Finally, as for the disscussion part in this paper, the paper lacks effective summaries of the existing literatures, and the conclusion fails to reflect the superiority of this paper.

Response: Thank you for pointing this out. The first part of the discussion now summarizes the results in a better way, page 4. We think that we have summarized the existing literature in the introduction, page 2, line 49-7,1 and in the discussion. However, we choose not to highlight all the texts of importance, it should be more difficult to read then. We have also revised the conclusion. Page 15, line 486-490, to better reflect the superiority of the study.

Reviewer 3 Report

Comments and Suggestions for Authors

In the manuscript submitted to me for review entitled "A starch- and sucrose-reduced diet has similar efficiency as low FODMAP in IBS. A randomized non-inferiority studythe authors Bodil Roth, Mohamed Nseir, Håkan Jeppsson, Mauro D’Amato, Kristina Sundquist and Bodil Ohlsson present a study comparing the application of two types of dietary regimens: with low content of fermentable oligo-, di-, and monosaccharides and polyols (FODMAP) and starch- and sucrose-reduced diet (SSRD). The participants in the study are aged 18 - 70 years and were followed for a total of 6 months. A positive side of the study is the inclusion of individuals of different ages, rather than concentrating attention on only one age group. All participants gave their consent to be included in the study. The research was conducted in accordance with the declaration of Helsinki and approved by the Swedish Ethical Review Authority.

The methods involved in the study are well described and reproducible. The obtained results are presented in detail using 2 figures and 5 tables in the main text of the manuscript and 6 additional tables presented in the supplementary file. The conclusions drawn by the authors are a well-formed and accurate summary of the results presented.

To support their research, the authors used 39 references that presented information from studies published mostly in the past decade. Almost 2/3 of the total references are from the last 5 years, and nearly 1/5 of the total number are from the current year 2024. This proves that the topic of the application of different food diets and their benefits for human health is actively being developed in recent years and would be of interest to Nutrients readers. I did not notice any redundant self-citations, all the references used are appropriate and necessary for the preparation of the manuscript.

My remarks and recommendations to the authors are:

1. The research participants were selected at the age of 18 - 70 years. It would be good if the results were divided into age groups, for example 18 - 39; 40 - 59 and over 60 years. The physical response to the diet should also be largely determined by the age of the individuals.

2. On line 82, several diagnoses are mentioned, but with their medical codes. Isn't it better for the reader to spell out the entire diagnoses?

3. In Figure 2, the captions inside the figure are very small, and using a higher magnification degrades their quality. If possible let them grow a little and improve their quality.

4. 2 Supplementary Figures and 6 Tables are indicated in Supplementary Materials at the end of the manuscript. But only the tables are presented in the attached supplementary file. I may be missing something, but let the authors check where Figures S1 and S2 are presented, or supplement them.

Author Response

Dear Editor-in-chief and Reviewers

Manuscript ID: nutrients-3166947
Type of manuscript: Article
Title: A starch- and sucrose-reduced diet has similar efficiency as low
FODMAP in IBS. A randomized non-inferiority study

We are most grateful to the reviewers for taking their time to read the manuscript and their very constructive comments. In respect to these comments, we have been able to improve the manuscript. We have tracked all changes in yellow. We hope that our work now will be regarded as suitable for publication in the Nutrients.

Bodil Ohlsson, Professor, Senior Consultant of Gastroenterology and Hepatology

 Reviewer 3.

In the manuscript submitted to me for review entitled "A starch- and sucrose-reduced diet has similar efficiency as low FODMAP in IBS. A randomized non-inferiority study“ the authors Bodil Roth, Mohamed Nseir, Håkan Jeppsson, Mauro D’Amato, Kristina Sundquist and Bodil Ohlsson present a study comparing the application of two types of dietary regimens: with low content of fermentable oligo-, di-, and monosaccharides and polyols (FODMAP) and starch- and sucrose-reduced diet (SSRD). The participants in the study are aged 18 - 70 years and were followed for a total of 6 months. A positive side of the study is the inclusion of individuals of different ages, rather than concentrating attention on only one age group. All participants gave their consent to be included in the study. The research was conducted in accordance with the declaration of Helsinki and approved by the Swedish Ethical Review Authority.

The methods involved in the study are well described and reproducible. The obtained results are presented in detail using 2 figures and 5 tables in the main text of the manuscript and 6 additional tables presented in the supplementary file. The conclusions drawn by the authors are a well-formed and accurate summary of the results presented.

To support their research, the authors used 39 references that presented information from studies published mostly in the past decade. Almost 2/3 of the total references are from the last 5 years, and nearly 1/5 of the total number are from the current year 2024. This proves that the topic of the application of different food diets and their benefits for human health is actively being developed in recent years and would be of interest to Nutrients readers. I did not notice any redundant self-citations, all the references used are appropriate and necessary for the preparation of the manuscript.

My remarks and recommendations to the authors are:

  1. The research participants were selected at the age of 18 - 70 years. It would be good if the results were divided into age groups, for example 18 - 39; 40 - 59 and over 60 years. The physical response to the diet should also be largely determined by the age of the individuals.

Response: Thanks for this consideration. We have now divided the participants into these age groups, but age did not affect the response rate at any occasion in any of the response definitions. We have chosen only to show the p-values of responder rates defined as ≥50 scores, as that is the main result, page 12, line 328-331. We think it will be exhausting for the reader to show several unsignificant values.

  1. On line 82, several diagnoses are mentioned, but with their medical codes. Isn't it better for the reader to spell out the entire diagnoses?

Response: Agree. We have now added the entire diagnoses as well, page 3, line 110-112. We think it is important to keep the ICD codes, because those are the codes you use for research in the medical journals, and those codes varies from time to time.

  1. In Figure 2, the captions inside the figure are very small, and using a higher magnification degrades their quality. If possible let them grow a little and improve their quality.

Response: We have now revised Figure 2 with bigger letters in the axis, but numbers within the figure is not possible to change. The numbers inside the figure reflect the coding of outliers, so those are not important to read.

  1. 2 Supplementary Figures and 6 Tables are indicated in Supplementary Materials at the end of the manuscript. But only the tables are presented in the attached supplementary file. I may be missing something, but let the authors check where Figures S1 and S2 are presented, or supplement them.

Response: You are not missing anything. It is a mistake in the submission process. Now supplementary Figure 1 and 2 are added after all the supplementary tables in the same document “Supplementary Material”.

Round 2

Reviewer 2 Report

Comments and Suggestions for Authors

No additional comments.